# DACFL: Dynamic Average Consensus-Based Federated Learning in Decentralized Sensors Network

**DOI:** 10.3390/s22093317

**Published:** 2022-04-26

**Authors:** Zhikun Chen, Daofeng Li, Jinkang Zhu, Sihai Zhang

**Affiliations:** 1Department of Electronic Engineering and Information Science, School of Information Science and Technology, University of Science and Technology of China, No. 96 Jinzhai Road, Hefei 230026, China; zhikunch@mail.ustc.edu.cn (Z.C.); df007@mail.ustc.edu.cn (D.L.); jkzhu@ustc.edu.cn (J.Z.); 2PCNSS Laboratory, School of Information Science and Technology, University of Science and Technology of China, Hefei 230026, China; 3CAS Key Laboratory of Wireless-Optical Communications, School of Information Science and Technology, University of Science and Technology of China, Hefei 230026, China; 4School of Microelectronics, University of Science and Technology of China, Hefei 230026, China

**Keywords:** decentralized sensors network, dynamic average consensus, federated learning

## Abstract

Federated Learning (FL) is a privacy-preserving way to utilize the sensitive data generated by smart sensors of user devices, where a central parameter server (PS) coordinates multiple user devices to train a global model. However, relying on centralized topology poses challenges when applying FL in a sensors network, including imbalanced communication congestion and possible single point of failure, especially on the PS. To alleviate these problems, we devise a Dynamic Average Consensus-based Federated Learning (DACFL) for implementing FL in a decentralized sensors network. Different from existing studies that replace the model aggregation roughly with neighbors’ average, we first transform the FL model aggregation, which is the most intractable in a decentralized topology, into the dynamic average consensus problem by treating a local training procedure as a discrete-time series.We then employ the first-order dynamic average consensus (FODAC) to estimate the average model, which not only solves the model aggregation for DACFL but also ensures model consistency as much as possible. To improve the performance with non-i.i.d data, each user also takes the neighbors’ average model as its next-round initialization, which prevents the possible local over-fitting. Besides, we also provide a basic theoretical analysis of DACFL on the premise of i.i.d data. The result validates the feasibility of DACFL in both time-invariant and time-varying topologies and declares that DACFL outperforms existing studies, including CDSGD and D-PSGD, in most cases. Take the result on Fashion-MNIST as a numerical example, with i.i.d data, our DACFL achieves 19∼34% and 3∼10% increases in average accuracy; with non-i.i.d data, our DACFL achieves 30∼50% and 0∼10% increases in average accuracy, compared to CDSGD and D-PSGD.

## 1. Introduction

The unprecedentedly growing intelligent devices with smart sensors are providing a vast amount of privacy-sensitive data, which are usually related to the device owners. According to the General Data Protection Regulation (GDPR) [1], how to utilize these data in a privacy-preserving way has become a critical issue in the smart sensors network. To this end, federated learning (FL) [2,3] advocates training the machine learning model and storing the data locally, uploading only the parameters to a central parameter server (PS) for model fusion. However, there are defects in FL because of relying on a centralized topology (Figure 1a). For example, the PS iteratively synchronizes multiple local models from user devices and sends back the result to them, which leads to an extremely imbalanced communication burden of the sensors network. In detail, multiple devices communicate with the PS concurrently, so the communication traffic jam is likely to happen to the PS, especially in a sensors network where the bandwidth may be usually low. What is worse, if the PS suffered a single point of failure, the FL would be paralyzed.

To alleviate the bottlenecks aroused by centralized topology, an intuitive idea is to facilitate FL in a decentralized topology without a PS (Figure 1b). Hopefully, existing studies on device-to-device (D2D) communication have conferred the communication ability in a decentralized sensors network [4,5,6]. Therefore, we argue that it is not only important but also applicable for a sensors network to apply FL in a decentralized way. Actually, there have already been some studies investigating FL in a decentralized topology (refer to Section 2.2). Apart from them, references [7,8] are the two most similar studies to our work. In [7], a consensus-based distributed SGD (CDSGD) for collaborative deep learning over a fixed (time-invariant) topology is proposed, which enables data parallelization and decentralized computation. However, there are two main limitations with CDSGD. (i) It requires a uniform interaction matrix of which the elements are identical; (ii) It assumes independent and identically distributed (i.i.d) data over all devices. Therefore, the CDSGD becomes infeasible for decentralized sensors network where a time-varying topology and non-i.i.d data are usually common occurrences. In [8], a decentralized parallel SGD (D-PSGD) is studied on a fixed decentralized ring topology. The D-PSGD is also inapplicable for a decentralized sensors network because there is no physical PS to perform an additional network-wide model average which is explicitly required in D-PSGD. Imagine, if all devices are required to perform a network-wide average, it would inevitably result in unacceptable communication congestion. Besides, a fixed ring topology in D-PSGD also encounters the same limitation (i) as CDSGD.

In this paper, we aim to facilitate FL in a more generic decentralized sensors network, involving densely and sparsely connected, as well as time-invariant and time-varying topology, with i.i.d and non-i.i.d data over user devices, while ensuring the consistency across users’ models as much as possible (Although there are studies about personalized FL which obtains personalized final models [9,10,11], they are beyond the scope of this paper). To this end, we propose a Dynamic Average Consensus-based Federated Learning (DACFL). Our insights are with three folds. First, we transform the model aggregation of FL into a dynamic average consensus problem. Specifically, user devices are connected through an undirected graph denoted by a doubly stochastic and symmetric matrix (also called a mixing matrix). The model parameters of each device in the training procedure are regarded as a discrete-time series. In this way, the FL model aggregation, whose objective is to generate a global average model, fits well the dynamic average consensus, whose goal is to estimate the global average of all reference inputs. Second, we apply the first-order dynamic average consensus (FODAC) [12] to approximate the average model, which solves the model aggregation in a decentralized way while ensuring consistency across different models. Third, to improve the performance on non-i.i.d data, each device uses its neighborhood weighted average model as its next-round model initialization, which prevents the possible local over-fitting problem during the training procedure. For a better understanding, we summarize the difference between DACFL, CDSGD, and D-PSGD in Table 1. In detail, instead of replacing roughly the model aggregation with neighbors’ model average, our DACFL applies FODAC to estimate the average model of all users, which ensures the model consistency; when compared to CDSGD, our DACFL is superior in time-varying topology, sparse topology, and non-i.i.d data; when comparing to D-PSGD, our DACFL requires no additional network-wide model average which avoids communication congestion.

The contributions of this paper are summarized as follows:This paper devises a new decentralized FL implementation coined as DACFL, which applies to a more generic decentralized sensors network topology while ensuring consistency across different users. Unlike CDSGD and D-PSGD roughly replacing the model aggregation with neighbors’ average, the DACFL treats each device’s local training as a discrete-time process and applies FODAC to estimate the average model, through which the devices can obtain a near-average model in the absence of PS during the training procedure.We provide a basic theoretical convergence analysis of DACFL with some assumptions. The numeric result offers a convergence guarantee of DACFL and reveals a positive correlation of the convergence speed to the learning rate and a negative correlation to the topology size. Specific experimental results also support our analysis.A line of experiments on public datasets show that our DACFL outperforms CDSGD and D-PSGD w.r.t Average of Acc and Var of Acc in most cases.

## 2. Related Works

In this section, we first provide a brief introduction to FL and then summarize existing studies about decentralized FL implementations and about dynamic average consensus.

### 2.1. Federated Learning

As per [3], FL can be categorized into Horizontal Federated Learning (HFL) [2,13], Vertical Federated Learning [14,15,16] and Federated Transfer Learning [17,18,19] based on the distribution characteristics of the data. Throughout this paper, we focus only on the HFL. In what follows, we present the basic concept of HFL.

In HFL, a distributed training model is executed by a number of devices that share local model updates with a central PS who aggregates these updates to build a global model. Generally, an FL scenario consists of two main phases, local update, and global aggregation. In the local update phase, devices compute the gradients to minimize the underlying loss function using their local data. While in the global aggregation phase, the PS collects model updates from devices, aggregates them to form a global model, and sends back the global model to devices for their next training.

Formally, suppose there is a subset of devices C⊆N selected by the PS at training epoch t≤T. Each device c∈C keeps a local dataset $D_{c}=\left\{X_{c},Y_{c}\right\}$, where $X_{c}\in R^{\left|D_{c}\right|\times d}$ represents the feature space of device *c*’s training data and $Y_{c}\in R^{\left|D_{c}\right|\times m}$ is the associated label space. Let $\ell \left(\omega ;x_{i},y_{i}\right)$ denote the loss function of data sample $x_{i}$, where ω is the model parameters, then the local loss of device *c* over training dataset $D_{c}$ can be expressed as
(1)$$f_{c}\left(\omega \right)=\frac{1}{\left|D_{c}\right|}\sum_{i\in D_{c}}\ell \left(\omega ;x_{i},y_{i}\right).$$

Then the global loss across all devices can be given as
(2)$$f\left(\omega \right)=\sum_{c=1}^{\left|C\right|}\frac{\left|D_{c}\right|}{\left|D\right|}f_{c}\left(\omega \right),$$
where $D={⋃}_{c}D_{c}$ represents for the whole training dataset over devices subset *C* and $\left|D\right|=\sum_{c=1}^{\left|C\right|}\left|D_{c}\right|$ denotes the total number of the data samples.

To solve the above-distributed optimization problem, an incomplete list of studies have offered their solutions [2,20,21,22,23,24,25,26]. In [2], the FederatedAveraging (FedAvg) is first advocated to combine local stochastic gradient descent (SGD) on each device with a server that performs model averaging. To address the communication bottleneck and the scalability of FL, a Federated Learning method with Periodic Averaging and Quantization (FedPAQ) is proposed [20], which consists of server periodic model averaging, partial device participation, and quantized message passing. In [21], a secure aggregation framework Turbo-Aggregate reduces the model aggregation overhead from O(N${}^{2}$) to O(NlogN) by employing a multi-group circular strategy with additive secret sharing and novel coding techniques. To reduce the up-link communication overhead and improve the performance on non-i.i.d data, a Semi-Federated Learning (Semi-FL) framework divides users into multiple clusters and uploads only the cluster heads’ models to the server [22]. By carefully designing an in-cluster sequential training manner, Semi-FL improves the performance on non-i.i.d data. Similar to Semi-FL, ref. [23] also divides users into clusters and devises a client-edge-cloud Hierarchical Federated Learning (Hier-FL), which reduces the costly communication with the cloud. In [24], a layers-wise Federated Matched Averaging (FedMA) is proposed for convolutional neural networks (CNNs) and long-short-term memory (LSTM) to address the data heterogeneity. In [25], the authors propose Federated Learning Based on a SPADE MAS (FLaMAS), which designs a multi-agent system to enable flexibility and dynamism in FL. In [26], a Federated Learning-Based Graph Convolutional Network (FedGCN) is proposed to process non-Euclidean data. All the above studies address the FL from a theoretical or practical point of view. However, they all rely on a centralized topology where a central PS is required to execute the global aggregation, which poses challenges including imbalanced communication burden and the single point of failure when applied to a sensors network. For a more comprehensive study of FL, please refer to [3,27,28,29,30,31].

### 2.2. Decentralized Implementation of Federated Learning

There have already been some studies enabling FL into a decentralized topology. A fully decentralized FL framework is proposed in [32], where users take a Bayesian-like approach to iterate and aggregate the beliefs of their one-hop neighbors and collaboratively estimate the global optimal parameter. In [33], a peer-to-peer approach, BrainTorrent is proposed targeting towards medical applications in which all clients are pair-wisely connected and update models by checking the local model version with the latest model version over the network. However, references [32,33] do not afford sufficient flexibility for users to manipulate compute-graph or node-level data sharing preferences. To this end, a universal framework, Scatterbrained, is therefore proposed [34]. In [35], a gossip communication protocol based on SGD, GossipGraD, is designed for scaling deep learning on large-scale systems without a PS, which reduces overall communication complexity and also enables asynchronous communication. Similarly, references [36,37,38] also design decentralized FL based on gossip protocol. In [36], the Combo is designed based on the gossip protocol and a model segmentation level synchronization mechanism, which is then extended to a bandwidth aware solution by greedily choosing the bandwidth-sufficient worker to reduce the transmission delay, called BACombo [37]. Furthermore, an experimental study compares gossip learning with FL and finds gossip learning is comparable to FL [38]. In addition to the gossip protocol, blockchain also assists the decentralized implementation of FL [39,40,41,42,43,44,45]. In [39], a blockchain-enabled FL (FL-Block) scheme enables the autonomous machine learning without any centralized authority to maintain the global model and coordinates by using a Proof-of-Work consensus mechanism of the blockchain, which improves the privacy issue and insufficient performance of fog computing. A crowdsourcing framework, CrowdSFL, where users can implement crowdsourcing with less overhead and higher security is proposed by combining FL with blockchain [40]. In [41], the BFLC framework uses blockchain to store the global model and exchange the local model update. In [45], a blockchain-assisted decentralized FL (BLADE-FL) is developed with an upper bound on the global loss function, based on which the authors optimize the computing resource allocation and explore the impact of lazy clients. In [42], a decentralized paradigm for big data-driven cognitive computing is developed by using a blockchain-enabled FL to introduce an incentive mechanism to solve the data island problem with privacy protection. An overview of the fundamentals of FL and blockchain is presented in [43]. The authors also propose the FLchain in mobile-edge computing networks by integrating FL with blockchain. In [44], the authors propose an innovative FL with asynchronous convergence (FedAC) considering a staleness coefficient by using a blockchain network to aggregate the global model, which avoids real-world issues such as interruption by abnormal local device training failure, dedicated attacks, etc.

Based on the above illustration, existing decentralized FL (DFL) implementations can be summarized into two main categories, DFL based on gossip protocol and DFL based on blockchain. Our solution differentiates these two categories from the following perspectives. (i) The methods are different. Unlike existing studies using gossip protocol or blockchain to accomplish the model aggregation in a decentralized topology, we first transform the model aggregation problem into a dynamic average consensus problem and then employ the FODAC to approximate the average model, thus tackling the model aggregation in a decentralized topology. (ii) The results achieved are different. Compared to DFL based on gossip protocol which usually requires a pair-wisely connected topology, our DACFL is robust to a more generic topology, not necessarily pair-wisely connected or even sparsely connected. The only prerequisite is a symmetric and doubly stochastic mixing matrix. As a result, our solution reduces the overall communication burden. While when compared to DFL based on blockchain, which is usually with high computational complexity, our solution is more less computationally complex. Therefore, the DACFL is more practically feasible when applied to a decentralized sensors network where the devices are usually with sparing capabilities of communication and computation.

### 2.3. Dynamic Average Consensus

The dynamic average consensus problem is referred to as the problem in which a set of autonomous agents aims to track the average of individually measured time-varying signals by local communication with neighbors. Existing papers have studied the dynamic average consensus problem regarding the continuous-time reference inputs. In [46], the authors use standard frequency-domain techniques and show that their algorithm can track the average of ramp reference inputs with zero steady-state error. In the context of input-to-state stability, the authors [47] show that a proportional dynamic average consensus algorithm can track with bounded steady-state error the average of bounded reference inputs with bounded derivatives. In [48], the authors propose a dynamic consensus algorithm and apply it to design consensus filters. Their algorithm can track with some bounded steady-state error the average of a common reference input with a bounded derivative. In [49], the authors further assume that agents know the nonlinear model, which generates the time-varying reference function. In [50], the $l_{1}$-regularized $H_{\infty }$ filtering is introduced to solve the estimation problem. In [51], a tracking algorithm for sparse and dynamic underwater sensor networks based on particle filter (TASD) is proposed to improve the slow convergence rate and low filtering accuracy of the traditional particle filter. While for the dynamic average consensus problem regarding the discrete-time reference inputs, reference [12] proposes a class of discrete-time dynamic average consensus algorithms and analyzes their convergence properties. The algorithms can track a class of time-varying reference inputs, including polynomials, logarithmic-type functions, periodic functions, and other functions whose *n*-th-order differences are bounded, for n≥1, with zero or sufficiently small steady-state error.

For our decentralized FL implementation in this paper, we employ the first-order dynamic average consensus (FODAC) algorithm [12] (see Algorithm 1) for each device to track the average model during the training procedure in the absence of central PS.
**Algorithm 1:** First-order dynamic average consensus [12].
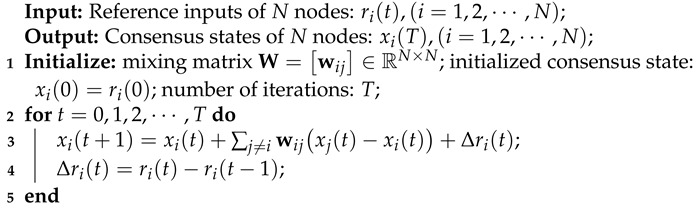


## 3. System Model and Problem Formulation

This section first provides the node model and communication model, then formally constitutes the decentralized FL implementation as a minimization optimization problem.

### 3.1. Node Model

A node model refers to a device (In the rest of the paper, we no longer distinguish user, device, and node), which contains a local dataset and a local model. Suppose there are *N* nodes in a decentralized topology, labeled by i∈V={1,2,⋯,N}. The local dataset on the *i*-th node is $D_{i}$, and the whole dataset is $D=D_{1}\cup \cdots \cup D_{N}$, where $D_{i}\cap D_{j}=\emptyset$ if i≠j. The local model of node *i* at round *t* is represented by $\omega_{i}^{t}$. Generally, the models on all nodes are required to be structural identical and initialized by the same parameters, i.e., $\omega_{1}^{0}=\omega_{2}^{0}=\cdots =\omega_{N}^{0}=\omega^{0}$. In the local update phase, each node trains its $\omega_{i}^{t}$ based on $D_{i}$ and generates a discrete-time series $\omega_{i}=\left\{\omega_{i}^{0},\omega_{i}^{1},\cdots ,\omega_{i}^{T}\right\}$. We denote ${\bar{\omega }}^{t}=\frac{1}{N}\sum_{i=1}^{N}\omega_{i}^{t}$ the average model over *N* nodes at round *t*. Therefore, the global aggregation phase aims to complete the ${\bar{\omega }}^{t}$. In the following part, we provide an estimation method to approximate the ${\bar{\omega }}^{t}$ in the absence of a PS.

### 3.2. Communication Model

A communication model refers to two rules that govern the information exchange between all nodes. (i) A connectivity rule ensuring that the information of each node influences the information of any other nodes; (ii) A rule on connection weights that a node uses when combing its own information with the information received from its neighbors. In practice, the connection weights may be affected by the distance or the channel interference, e.g., a longer distance or a more severe interference corresponds to smaller connection weights and vice versa.

The decentralized topology at round *t* is represented as an undirected graph $G\left(t\right)=\left(V,E\left(t\right)\right)$, where *V* is a node set and E(t)⊂V×V is an edge set. We call node *i* and *j* neighbors to each other if (i,j)∈E(t), which indicates node *i* and *j* are enabled one-hop communication. A connected graph is required such that the information of node *i* can influence the information of any other nodes directly or indirectly. For simplicity, we use a mixing matrix $W\left(t\right)=\left[w_{ij}\left(t\right)\right]\in R^{N\times N}$ to denote the graph with $w_{ij}\left(t\right)$ representing for the connection weights demonstrated in the above rule (ii) between node *i* and *j*, where
(3)$$\left\{\begin{array}{c} 0<w_{ij}\left(t\right)<1,\mathrm{if}\;\left(i,j\right)\in E\left(t\right)\\ w_{ij}\left(t\right)=0,\mathrm{if}\;\left(i,j\right)\notin E\left(t\right) \end{array}.\right.$$

Here W(t) is required to be symmetric and doubly stochastic, i.e., W(t)1=1, $1^{T}W\left(t\right)=1^{T}$.

### 3.3. Problem Formulation

Based on the above models, an important remaining problem is to execute the global aggregation phase without PS. In the following, we transform the global aggregation in a decentralized topology into a dynamic average consensus problem. Specifically, we treat the local update phase as a discrete-time process and take the intermediate model parameters $\omega_{i}=\left\{\omega_{i}^{0},\omega_{i}^{1},\cdots ,\omega_{i}^{T}\right\}$ as the reference inputs on node *i*. Then, the global aggregation whose goal is to obtain the average model can be transformed into a dynamic average consensus problem, of which the objective is to track the average of the reference inputs over all nodes,
(4)$$\min_{x^{t}}{∥x^{t}-{\bar{\omega }}^{t-1}1∥}_{2}^{2}.$$

Here $x^{t}=\left[x_{1}^{t};x_{2}^{t};\cdots ;x_{N}^{t}\right]$ is a vector denoting all the estimations on *N* nodes at round *t*. In this paper, we employ the FODAC (Algorithm 1) to solve the problem in Equation (Equation 4). Besides, as is shown in (Equation 2), the objective of FL is to minimize the global loss. So, we formally combine Equations (Equation 2) and (Equation 4) and summarize the objective of the decentralized FL as
(5)$$\left\{\begin{array}{c} \min_{\omega }f(\omega ):=\sum_{i=1}^{N}\frac{\left|D_{i}\right|}{\left|D\right|}f_{i}\left(\omega \right)\\ \min_{x^{t}}{∥x^{t}-{\bar{\omega }}^{t-1}1∥}_{2}^{2} \end{array}.\right.$$

Especially, if each node holds the same number of training data, Equation (Equation 5) can be further expressed as
(6)$$\left\{\begin{array}{c} \min_{\omega }f(\omega ):=\frac{1}{N}\sum_{i=1}^{N}f_{i}\left(\omega \right)\\ \min_{x^{t}}{∥x^{t}-{\bar{\omega }}^{t-1}1∥}_{2}^{2} \end{array}.\right.$$

In Section 4, we design a DACFL algorithm to solve this problem.

## 4. Methods

In this section, we first construct the symmetric doubly stochastic matrix, which is further used to connect multiple user devices. Then, we briefly introduce the first-order dynamic average consensus and apply it to design the DACFL for implementing FL in a decentralized sensors network. Figure 2 shows an overview of our solution, which is explained in Section 4.3.

### 4.1. Construct a Symmetric Doubly Stochastic Matrix

As illustrated in Section 3, the mixing matrix is essential for our decentralized communication model. Therefore, it is important to construct the mixing matrix beforehand. Actually, a very simple doubly stochastic and symmetric matrix could be designed as $W:={\left[\frac{1}{n}\right]}^{n\times n}$. However, to get a more generic mixing matrix, we use the Sinkhorn–Knopp algorithm [52] to design the doubly stochastic and symmetric matrix in this paper.

### 4.2. First-Order Dynamic Average Consensus

As in Section 3.3, by treating the intermediate models during the local update phase as the discrete-time reference inputs, the average model can be approximated by a dynamic average consensus algorithm. In this paper, we use the FODAC to approximate the average model in our DACFL, which has been proved to track the average state with either a zero steady-state error or an upper-bounded steady-state error [12]. Algorithm 1 briefly shows thepseudo-codee of FODAC.

### 4.3. Dynamic Average Consensus-Based Federated Learning

In Equation (Equation 5), the objective of decentralized FL implementation is to simultaneously minimize a global loss and solve a dynamic average consensus problem. For the former sub-problem, a distributed stochastic gradient descent can be used here-in, while for the latter sub-problem, we employ the FODAC to solve it. By combining the distributed gradient descent and the FODAC, we devise an algorithm, Dynamic Average Consensus-based Federated Learning (DACFL). Figure 2 shows an overview of the DACFL, which is constituted of three main stages. In stage I, multiple user devices are connected through the mixing matrix, which can be constructed by the methods introduced in Section 4.1. In stage II, each device parallel performs the training procedure. In stage III, the estimated average models are output as the final result. Four main steps constitute the stage II of DACFL: (i) each device trains its own model using its local data; (ii) each device computes a neighborhood weighted average model $\omega_{i}^{t^{\prime }}$ by exchanging its intermediate model with its neighbors; (iii) each device performs the FODAC to track the average model. (iv) each device takes the neighborhood weighted average model as its next-round initialization. More specifically, step (i) can also be referred to as the local update phase in FL. In step (ii), the neighborhood weighted average model is further used as the device’s next-round initialization in step (iv) (line 6, Algorithm 2), which is empirically demonstrated more robust to sparse topology and non-i.i.d data as it to some extent prevents the local over-fitting. In step (iii), we employ the FODAC to estimate the average model of all users in a decentralized way, which helps to handle the global aggregation phase without a central PS. The pseudo-code of the DACFL training procedure is summarized in the Algorithm 2.
**Algorithm 2:** Dynamic Average Consensus-based Federated Learning.
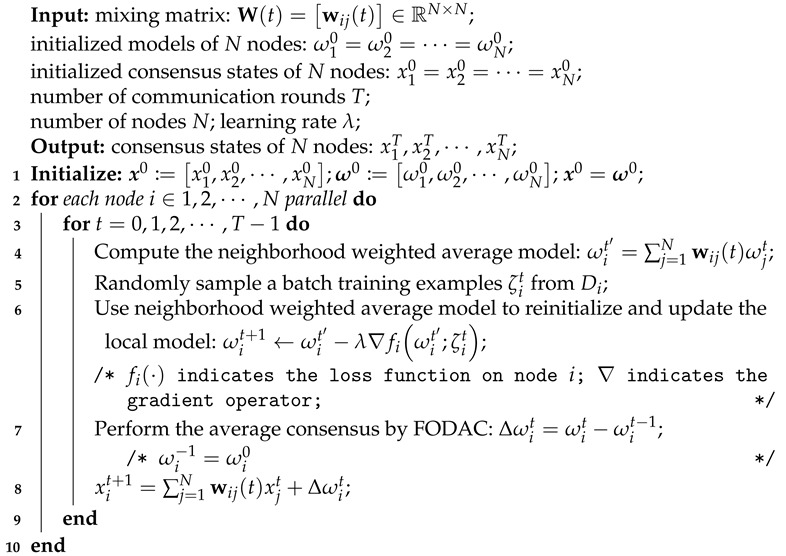


## 5. Convergence Analysis

In this section, we provide a basic theoretical proof of our solution with i.i.d data. Without loss of generality, each user is assumed to hold the same number of training data such that the loss function f(x) can be denoted as
(7)$$f(\omega ):=\frac{1}{N}\sum_{i=1}^{N}f_{i}\left(\omega \right).$$

To complete the analysis, we make the following assumptions.

**Assumption** **1** (L-Smooth)**.**
*Each loss function $f_{i}\left(\omega \right)$ is L-smooth such that*

(8)
$$f_{i}\left(v\right)\leq f_{i}\left(w\right)+{\left(v-w\right)}^{T}\nabla f_{i}\left(w\right)+\frac{L}{2}{∥v-w∥}_{2}^{2}.$$



In decentralized FL, each user holds its local dataset $D_{i}$ and gradient $\nabla f_{i}\left(\omega \right)$. To declare the user-wise gradients $\nabla f_{i}\left(\omega \right)$, we have Assumptions 2 and 3 on the premise of i.i.d data.

**Assumption** **2** (Bounded Gradients)**.**
*For each user i and a randomly sampled batch data $\zeta_{i}$, there exists an upper bound G>0 such that*

(9)
$$E_{\zeta_{i}∼D_{i}}{∥\nabla f_{i}\left(\omega ;\zeta_{i}\right)∥}^{2}\leq G^{2}.$$



**Assumption** **3** (Uniform Gradient First-order Difference)**.**
*At any round t, define*

$$\Delta g_{i}^{t}=\nabla f_{i}\left(\omega^{t^{\prime }};\zeta_{i}\right)-\nabla f_{i}\left(\omega^{t-1^{\prime }};\zeta_{i}\right)=g_{i}^{t}-g_{i}^{t-1}$$

*as the first-order difference of gradient, this paper assumes an i.i.d data distribution across all users for the convergence analysis, such that*

(10)
$$E\left[\Delta g_{i}^{t}\right]=E\left[\Delta g_{j}^{t}\right]=\Delta g^{t},\forall i,j,1\leq i,j\leq N.$$



**Assumption** **4** (Bounded First-order Differences of Model Parameter)**.**
*At any round t, there is a constant θ>0 ensuring an upper bound of each user’s model parameters such that*

(11)
$${∥\omega_{i}^{t}-\omega_{i}^{t-1}∥}^{2}\leq \theta^{2}.$$



In practice, a sufficiently small learning rate λ guarantees this assumption. Following the Assumption 4, we have
(12)$$\Delta \omega_{\max}^{t}-\Delta \omega_{\min}^{t}\leq \kappa ,$$
where κ is an upper bound related to θ. We also define
$$\begin{array}{cc} \Delta \omega_{i}^{t}:= & \omega_{i}^{t}-\omega_{i}^{t-1},\\ \Delta \omega_{\max}^{t}:= & \max_{i=1,2,\cdots ,N}\Delta \omega_{i}^{t},\\ \Delta \omega_{\min}^{t}:= & \min_{i=1,2,\cdots ,N}\Delta \omega_{i}^{t}. \end{array}$$

The above (Equation 12) ensures the FODAC tracks the average of the time-varying model parameters $\omega_{i}^{t}$ with a sufficiently small steady-state error. Detailed proof can be found in [12]. Following the above assumptions, we present the convergence rate of DACFL in Theorem 1.

**Theorem** **1.**
*Following the aforementioned assumptions, we have the average expected squared gradient norm following*

(13)
$$\begin{array}{c} \begin{array}{cc} \frac{1}{T}\sum_{t=0}^{T-1}{∥\nabla f({\bar{\omega }}^{t})∥}^{2}\; & \leq \frac{2}{\lambda T}E\left(f\left({\bar{\omega }}^{0}\right)-f\left({\bar{\omega }}^{T}\right)\right)+G^{2}+\frac{\theta^{2}}{\lambda^{2}}+\frac{L\theta^{2}}{\lambda }\\ \; & \leq \underset{:C_{0}}{\underbrace{\frac{2}{\lambda T}\left(f\left({\bar{\omega }}^{0}\right)-f^{*}\right)}}+\underset{:C_{1}}{\underbrace{G^{2}+\frac{\theta^{2}}{\lambda^{2}}+\frac{L\theta^{2}}{\lambda }}}, \end{array} \end{array}$$

*where $f^{*}$ denotes the minimum loss value of f(x).*


In (Equation 13), the average squared gradient norm is bounded by $C_{0}+C_{1}$, where the $C_{0}$ gradually tends to 0 when training iteration *T* increases. In other words, the average squared gradient norm is bounded by a learning rate related term $C_{1}$ when T→+∞. So, if ${\bar{\omega }}^{t}$ is regarded as the solution of *f*, the convergence is guaranteed. Besides, the final output of DACFL $x^{T}=\left[x_{1}^{T},x_{2}^{T},\cdots ,x_{N}^{T}\right]$ tracks the ${\bar{\omega }}^{T}$ with a sufficiently small error, hence providing a convergence guarantee for DACFL. For detailed proof of Theorem 1, please refer to the Appendix A.

## 6. Experiments and Performance Evaluation

In this section, we first declare the experimental setup and then evaluate the performance of DACFL with different topology and data allocations.

### 6.1. Experimental Setup

#### 6.1.1. Datasets, Allocation, Topology, and Neural Network Structure

(1) Datasets: Three public datasets including MNIST, Fashion-MNIST (FMNIST) and CIFAR-10 are used for our performance validation in this paper [53,54,55]. (i) MNIST: a dataset includes 70,000 images of hand-written digits, total of 10 classes, with a training set consisting of 60,000 examples and a test set consisting of 10,000 examples, respectively. (ii) Fashion-MNIST: a similar dataset comprises 28 × 28 grayscale images of 70,000 fashion products from 10 categories. (iii) CIFAR-10: a dataset consists totally of 60,000 color images with three RGB channels, which can be classified into 10 classes. The training set has 50,000 examples and the test set has 10,000 examples.

(2) Allocation: We design two ways for data allocations. (i) i.i.d: each user is assigned the same number of training samples with a uniformly random distribution over 10 classes. (ii) non-i.i.d: the training set is sorted by labels first and then divided into multiple shards with the same training data; each user samples 2 shards without replacement.

(3) Topology: Several topology from different perspectives are designed. (i) Time-varying and time-invariant: for the time-invariant topology, we initialize the mixing matrix and keep it unchanged during the training process; for the time-varying topology, we reconstruct the mixing matrix every 10 training rounds. (ii) sparse and dense connectivity: for the sparse topology, half elements of the mixing matrix are 0 (ψ = 0.5); while for the dense topology, all elements in the mixing matrix are non-zeros (ψ = 1.0).

(4) Neural Network Structure: We use the same CNN structure for MNIST and FMNIST which contains two 5 × 5 convolutional layers (each layer is followed with a batch normalization and 2 × 2 max pooling), a fully connected layer with ReLu activation and a final softmax output layer. For CIFAR-10, we use a CNN consisting of two convolutional layers (each layer is followed with batch normalization, ReLu activation, and 2 × 2 max-pooling), two fully connected layers with ReLu activation, and a final softmax output layer.

#### 6.1.2. Baselines and Performance Metrics

We compare DACFL with a centralized FL method, FedAvg [2], and another two decentralized methods, CDSGD [7] and D-PSGD [8]. For FedAvg, all users participate in each training round. For CDSGD and D-PSGD, the mixing matrix and other hyper-parameters are consistent with DACFL. Two metrics, including Average of Acc and Var of Acc are defined to indicate the performances. In detail, each user’s trained model (or estimated model in DACFL) is separately test, then an averaged result of all users’ test accuracy is used as Average of Acc and the variance over all users’ test accuracy is used as Var of Acc. For FedAvg and D-PSGD, the final output is a only global model; therefore, the Average of Acc is actually the same as the test accuracy and the Var of Acc is always 0.

We carried out the experiments on a Ubuntu 18.04 computer with 4 RTX 2080Ti GPU cards. All the baselines and our proposed solution are implemented by Python 3.8 and Pytorch 1.8.2 with CUDA 10.2, and we use MATLAB to visualize the result. Unless otherwise specified, some important parameters are set as Table 2.

### 6.2. Why Choose FODAC? A Numerical Perspective

A specific numerical experiment is designed in this section to clarify the benefit of FODAC. Specifically, we separately apply FODAC, CDSGD and D-PSGD to track the average of two types of discrete-time inputs under three different mixing matrices. (i) $R_{i}\left(t\right)=\sin\left(t\right)+{\left(\frac{1}{t}\right)}^{i}+t+i$, inputs with relatively large variance between each user; (ii) $R_{i}\left(t\right)=\sin\left(t\right)+{\left(\frac{1}{t}\right)}^{i}+t$, inputs with relatively small variance. Here $R_{i}\left(t\right)$ denotes the input of *i*-th user at time *t*, where $i\in \left\{1,2,\cdots ,10\right\}$, $t\in \left\{1,2,\cdots ,20\right\}$.

Three 10 × 10 mixing matrices are defined in this experiment. (i) sparse: ψ = 0.5, i.e., half elements are 0; (ii) dense: ψ = 1.0, i.e., all elements are non-zeros; (iii) uniform: all elements are 0.1. For CDSGD, we take roughly the neighborhood weighted average as the estimated value; for D-PSGD, the estimated value is the network-wide average on the estimation by CDSGD. While for FODAC, we take the consensus state (see line 3 in Algorithm 1) as the estimated result. Then, the absolute error can be computed by
(14)$$\mathrm{err}=\left|{\bar{R}}_{i}\left(t\right)-{\hat{R}}_{i}\left(t\right)\right|,$$
here ${\hat{R}}_{i}\left(t\right)$ is the estimation and ${\bar{R}}_{i}\left(t\right)$ is the average of inputs. Figure 3 shows a comparison of these three methods.

With large-variance inputs, because CDSGD roughly takes the neighbors’ average as the mean value, which becomes distorted with either large variance inputs or an ununiform mixing matrix. Therefore, a distinct error and variance in both sparse and dense mixing matrices in Figure 3 is observed. Comparatively, the only feasible result of CDSGD with small variance inputs or uniform mixing matrix also supports our suspect. While with small-variance inputs in Figure 3b, the FODAC still outperforms CDSGD from convergence speed in both sparse and dense mixing matrix. The D-PSGD is observed smaller error and variance than FODAC and CDSGD because it additionally carries out a network-wide average and takes this network-wide average as the mean value, which is exactly equal to the actual average value. However, such a network-wide average is practically infeasible when it comes to a decentralized sensors network where no physical PS is responsible for it. To sum up, the FODAC is more feasible than CDSGD and D-PSGD in approximating the average in a more generic decentralized topology; this actually motivates us to apply it into DACFL.

### 6.3. Performance on i.i.d Data

Figure 4 and Figure 5, respectively, show the performances with i.i.d data allocation in time-invariant and time-varying topology. The parameters are set following Table 2.

#### 6.3.1. Time-Invariant Topology

Figure 4 shows the result with i.i.d data in time-invariant topology. The following conclusions can be drawn from this result.

First, the DACFL outperforms D-PSGD and CDSGD in terms of Average Acc, albeit slightly inferior to FedAvg. From Figure 4a–c, the DACFL achieves 97%, 86%, 70% accuracy and 96%, 85%, 67% accuracy in a densely and sparsely connected topology, on three datasets, which is superior to the result of CDSGD with 93%, 64%, 18% accuracy and 68%, 51%, 20% accuracy, and the result of D-PSGD with 97%, 83%, 55% accuracy and 95%, 75%, 45% accuracy. The D-PSGD has higher accuracy than CDSGD because it additionally performs a model averaged over all users, which, however, leads to unacceptable communication congestion or even becomes practically infeasible when there is no physical PS in a decentralized sensors network. Nonetheless, due to the ununiform mixing matrix, its final accuracy is still slightly worse than FedAvg. As a deviation across different local models always exists in an ununiform mixing matrix, where the FODAC approximates the average model more effectively, which further leads to a superior accuracy for DACFL.

Second, the DACFL is more robust to the sparse topology. Specifically, the DACFL has 1%, 1%, 3% accuracy degradation on three datasets. While the D-PSGD degrades 2%, 8%, 10% accuracy, and the CDSGD degrades more than 10% accuracy. Because the FODAC is always effective as long as the mixing matrix satisfies the symmetric and doubly stochastic property, the DACFL is also feasible in sparse topology. In contrast, replacing roughly the average model with the neighbors’ average is very sensitive to the sparsity of the mixing matrix. Therefore, the CDSGD and D-PSGD show more serious accuracy degradation in sparse topology.

Third, the result of each user in DACFL is more consistent than that in CDSGD. Figure 4d–f show the variance of accuracy across different users. The variance in DACFL is observed to be smaller and more stable compared to CDSGD, which gradually tends to around 0 as the training progresses. This is because the FODAC is more effective in estimating the average model while the CDSGD replacing roughly the average model with the neighbors’ average may generate more diversified local models.

#### 6.3.2. Time-Varying Topology

Figure 5 presents the result on i.i.d data in time-varying topology. Because the time-varying mixing matrix is also ensured symmetric and doubly stochastic, the DACFL still outperforms D-PSGD and CDSGD. For example, in Figure 5b, the DACFL achieves an 87% accuracy better than the 84% accuracy of D-PSGD and 68% of CDSGD. Changing the topology does not affect the FedAvg, which performs better than other decentralized methods. Besides, the accuracy degradation caused by the topology sparsity becomes smaller for all decentralized methods. Especially, for D-PSGD on FMNIST (Figure 5b) and CIFAR-10 (Figure 5c), the average accuracy under a sparse topology is even greater than that under a dense topology. We suspect the randomness introduced by time-varying topology reduces the possibility of early over-fitting in a sparse topology. Finally, for the variance shown in Figure 5d–f, a similar result to that of a time-invariant topology is observed. The DACFL has smaller and more stable variance of accuracy than CDSGD in both densely and sparsely connected topology.

In summary, with i.i.d data, the DACFL is superior to CDSGD and D-PSGD in both time-varying and time-invariant topology.

### 6.4. Performance on Non-i.i.d Data

In this section, we test the performance of DACFL on non-i.i.d data and show the result in Figure 6 and Figure 7. The parameters are set following Table 2.

#### 6.4.1. Time-Invariant Topology

Figure 6 presents the result on non-i.i.d data in time-invariant topology. Comparing with results on i.i.d data, all methods have accuracy degradation on three datasets. This is because the non-i.i.d property leads to users’ local model divergence and early over-fitting. Since D-PSGD additionally performs a network-wide model average, it has higher accuracy than DACFL and CDSGD in MNIST and FMNIST (The result on CIFAR-10 is not counted here because all decentralized methods do not converge after 100 rounds). However, a network-wide model average usually results in an excessive communication overhead, especially when users are very dispersedly distributed in a large topology. In case of no network-wide model average, our DACFL outperforms CDSGD. Take the result in dense topology as an example; DACFL gets average accuracy of 86%, 70%, while CDSGD only achieves 58%, 40% on MNIST and FMNIST, respectively. It is the re-initialization by neighbors’ weighted average model which prevents the possible local over-fitting during the training procedure, thus improving the performance of DACFL in non-i.i.d data. Besides, the variance results in Figure 6d–f also shows the superiority of DACFL compared to CDSGD. Actually, the variance is larger than that of i.i.d data due to the non-i.i.d property. This is because that non-i.i.d would inherently lead to heterogeneous local models; therefore, CDSGD replacing the average model with the neighbors’ average would result in more diversified models across different users. However, the variance of DACFL still decreases and stabilizes as the training progresses due to the effectiveness of FODAC estimating the average model.

#### 6.4.2. Time-Varying Topology

Figure 7 presents the result on non-i.i.d data in time-varying topology. For the average accuracy in Figure 7a–c, similar accuracy degradation due to non-i.i.d data still exists when compared with the result on i.i.d data. Because the randomness introduced by time-varying topology possibly alleviates the local model over-fitting, our DACFL has a better performance than that of a time-invariant topology shown in Figure 6, especially on FMNIST and CIFAR-10. For the variance, similar result to Figure 6 happens, i.e., the variance of DACFL gradually decreases and tends to 0, which confirms the viability of DACFL in tracking the average model.

To sum up, the DACFL is also viable on non-i.i.d data (MNIST and FMNIST) under time-varying topology.

### 6.5. Convergence vs. Learning Rate and Topology Size

To find out how the learning rate and topology size affect our solution, the average test accuracy and average training loss on i.i.d MNIST with different learning rates and topology sizes are shown in Figure 8. Note only densely connected topology is considered in this part. Except for learning rate and topology size, other parameters in this experiment also follow Table 2.

#### 6.5.1. Performance vs. Learning Rate

Figure 8a,b show the test accuracy and training loss vs. different learning rates. From Figure 8a,b, a larger λ within the range 0.001≤λ≤0.01 leads to faster convergence. This is because a larger learning rate makes the loss function decreases with a larger step size, which leads to a faster convergence. However, this situation is the opposite when 0.05≤λ≤0.1, i.e., when λ increases from 0.05 to 0.1, the convergence speed and convergence result become worse, with a smaller average test accuracy and larger average training loss. It is because an excessive learning rate λ would lead to a larger upper bound of first-order difference of model parameter θ and thus cause a relatively larger upper bound of first-order difference κ in (Equation 12). Consequently, a larger steady-state error while using FODAC arises. So, λ=0.01 should be the best choice in this experiment, which gets a higher average test accuracy and lower variance while ensuring fast convergence.

#### 6.5.2. Performance vs. Topology Size

Figure 8c,d present the result with different topology size. In Figure 8c,d, as the size *N* grows, the convergence speed slows down and the final average accuracy declines. We suspect a larger topology size would lead to a larger deviation across users, which further poses a challenge when using FODAC to track the average model.

In summary, a proper learning rate can accelerate the convergence of DACFL. Although DACFL is robust to different topologies, a smaller size helps attain a better performance within limited training rounds.

## 7. Conclusions and Future Work

Over-reliance on centralized topology poses challenges, including imbalanced communication burden and possible single point of failure, for the application of FL in smart sensors network. For this reason, existing studies have proposed different methods like CDSGD and D-PSGD for implementing FL in a decentralized topology, which, however, have some deficiencies. Specifically, there exists distinct variance across users’ models in CDSGD, while the D-PSGD additionally requires a network-wide model average which is infeasible in practical decentralized sensors network due to no physical PS or unacceptable communication congestion. Therefore, this paper devises a new decentralized FL scheme, DACFL, aiming to implement FL in a decentralized sensors network while ensuring consistency across user devices. To this end, we first transform the most critical FL model aggregation in decentralized topology into a dynamic average consensus problem by treating the local training procedure as a discrete-time series. We then employ FODAC to track the average model across all users. We also provide a basic theoretical analysis of i.i.d data, which offers a convergence guarantee of our solution. Specific experiments on different public datasets verify the feasibility of DACFL in a more generic topology and declare the superiority of DACFL over CDSGD and D-PSGD.

Some issues need further investigation. First, a more communication-efficient method for DACFL is deserved because of each device in DACFL exchanges both estimation states and local models during the training process. This is especially important in a practical sensors network, usually with the insufficient capability of communication and computation. Two high-level ways are suggested to improve communication efficiency. One is to reduce the amount of data transmitted per communication round using a line of compression techniques; the other is to reduce the number of total communication rounds by speeding up the convergence. To reduce the computational burden, methods like network pruning and binary connection would help design a lightweight NN structure to reduce the computational burden. Second, an asynchronous DACFL deserves investigation for practical application. In this paper, we simply assume a synchronization across devices which would be, however, time-consuming with heterogeneous communication and computation capabilities of devices, because a faster device should wait for a slower device until it finishes its task before exchanging model information. Therefore, a staleness-aware asynchronous training mechanism needs to be carefully designed for DACFL in the future. Third, designing a halfway-drop and halfway-join-aware DACFL is worthy of future investigation. In a practical smart sensors network, some devices may be disconnected and reconnected during the training procedure, which would destroy the mixing matrix. Therefore, how to seamlessly reconstruct a symmetric and doubly stochastic matrix is very important. 

## Figures and Tables

**Figure 1 sensors-22-03317-f001:**
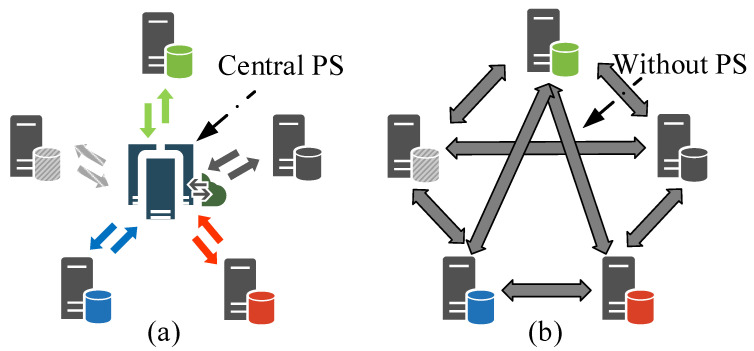
An overview of (**a**) centralized topology and (**b**) decentralized topology.

**Figure 2 sensors-22-03317-f002:**
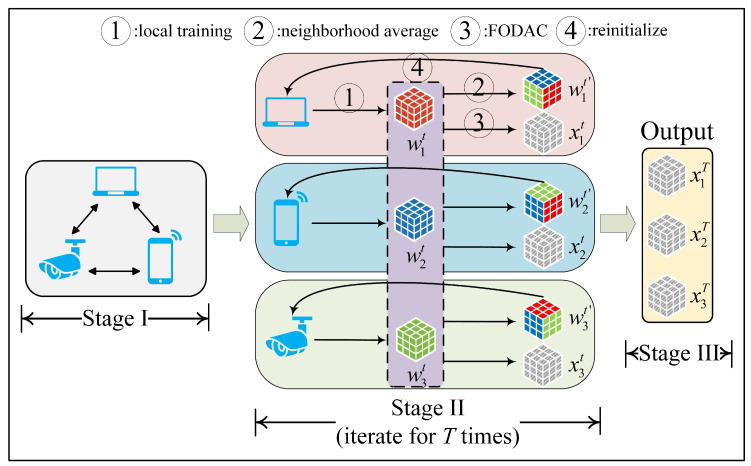
An overview of DACFL. Here a simple decentralized topology with 3 devices being pair-wisely connected is used as an example. Actually, the “neighborhood average” and “FODAC” in Stage II are carried out based on the models from a device’s neighbors, which contains not necessarily all models, but rely on a topology (not necessarily pair-wisely connected) instead.

**Figure 3 sensors-22-03317-f003:**
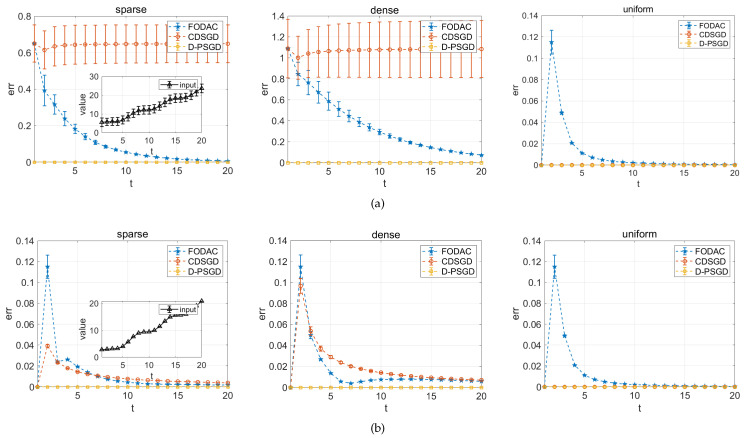
The result of approximating the average by different methods. (**a**) inputs with large variance; (**b**) inputs with small variance.

**Figure 4 sensors-22-03317-f004:**
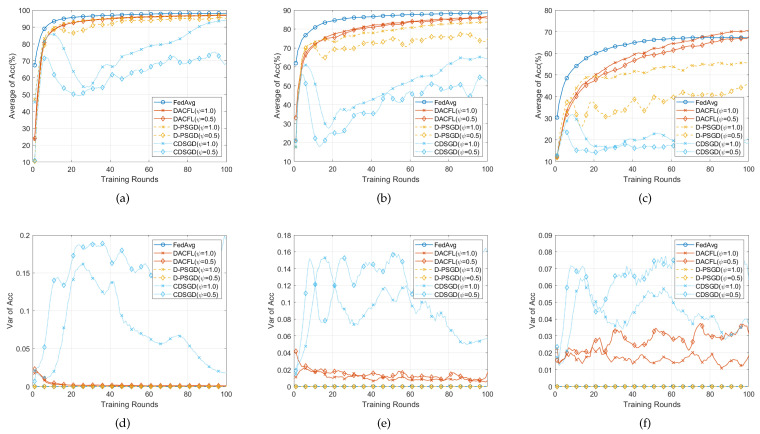
Performance comparison with i.i.d data and time-invariant topology. (**a**,**d**) on MNIST; (**b**,**e**) on FMNIST; (**c**,**f**) on CIFAR-10.

**Figure 5 sensors-22-03317-f005:**
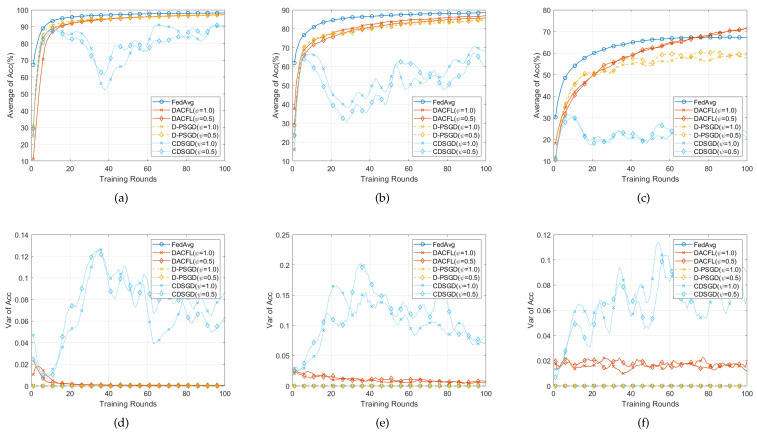
Performance comparison with i.i.d data and time-varying topology. (**a**,**d**) on MNIST; (**b**,**e**) on FMNIST; (**c**,**f**) on CIFAR-10.

**Figure 6 sensors-22-03317-f006:**
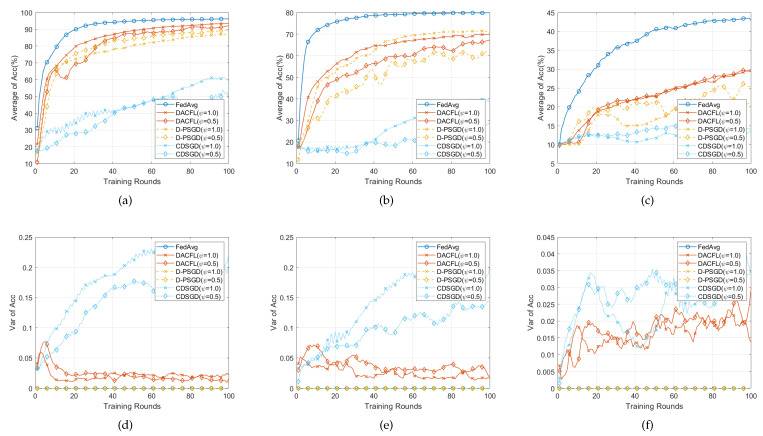
Performance comparison with non-i.i.d data and time-invariant topology. (**a**,**d**) on MNIST; (**b**,**e**) on FMNIST; (**c**,**f**) on CIFAR-10.

**Figure 7 sensors-22-03317-f007:**
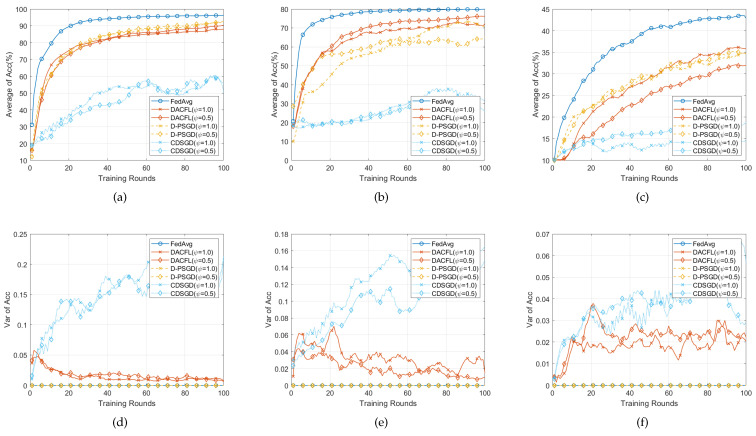
Performance comparison with non-i.i.d data and time-varying topology. (**a**,**d**) on MNIST; (**b**,**e**) on FMNIST; (**c**,**f**) on CIFAR-10.

**Figure 8 sensors-22-03317-f008:**
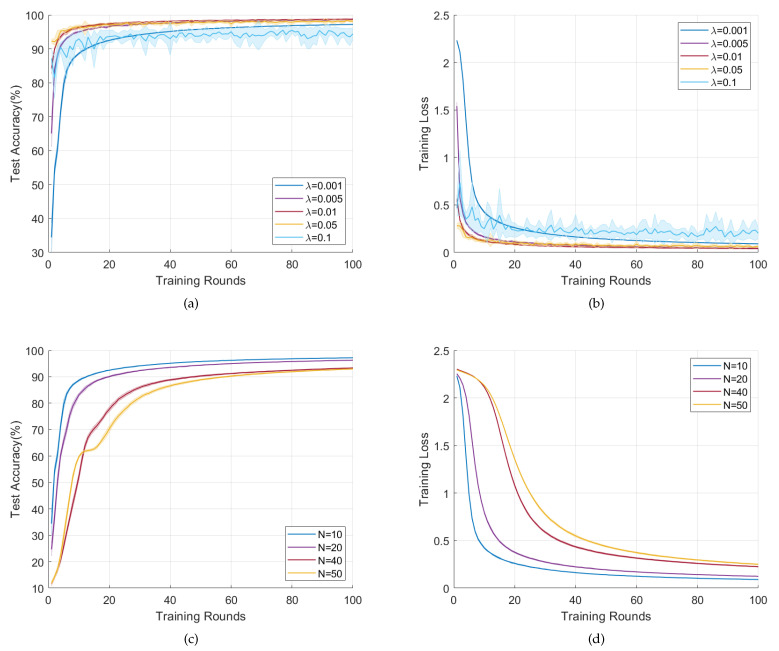
Performance vs learning rate and topology size. (**a**) Acc vs. λ; (**b**) Loss vs. λ; (**c**) Acc vs. N; (**d**) Loss vs. N.

**Table 1 sensors-22-03317-t001:** Comparison with CDSGD, D-PSGD. **✓** means enabled and **✗** means disabled in our result.

Solution	Model Aggregation	Network-Wide Average	Time- Invariant	Time- Varying	Dense	Sparse	i.i.d	Non-i.i.d
CDSGD [7]	replace by neighbors’ average	not required	**✓**	**✗**	**✓**	**✗**	**✓**	**✗**
D-PSGD [8]	replace by neighbors’ average	required	**✓**	**✓**	**✓**	**✓**	**✓**	**✓**
DACFL (ours)	by FODAC	not required	**✓**	**✓**	**✓**	**✓**	**✓**	**✓**

**Table 2 sensors-22-03317-t002:** Experimental parameters setting.

Parameter	Numeric Value
Number of nodes:	10
Training rounds:	100
Local batch size:	20
Local epoch:	1
Decaying for learning rate:	0.995
Loss function:	Cross Entropy
Learning rate:	MNIST/FMNIST: 0.001, CIFAR: 0.005

## Data Availability

Not applicable.

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
