# Peer review of "DACFL: Dynamic Average Consensus-Based Federated Learning in Decentralized Sensors Network"

_sensors, 2022, doi:10.3390/s22093317_

Round 1

Reviewer 1 Report

Reviewer's summary after reading the manuscript:

Multiple users can work together to train one global model using a centralized parameter server (PS) in a distributed learning system known as federated learning (FL). Traditional FL uses a central topology that will be paralyzed when the PS fails. Existing studies like CDSGD and D-PSGD allow FL in a decentralized topology to mitigate this single point failure. Some issues persist, such as large model divergence or a network-wide average need. Here, the authors propose the DACFL (Dynamic Average Consensus-based Federated Learning), which uses a symmetric and doubly stochastic matrix to allow each user to train his or her model using just his or her own data. The global average model is estimated using the first order dynamic average consensus (FODAC), which interprets the local training of each user as a discrete-time process. In addition, the authors present a foundational theoretical study of DACFL based on i.i.d data. D-PSGD and CDSGD are outperformed by DACFL in the majority of situations, proving the validity of DACFL in time-invariant and time-varying topology.

----------------------------------------

Dear authors, thank you for your manuscript. I enjoyed reading it. Presented are some suggestions to improve it:

(1) Please consider modifying the title of the manuscript to include the word "sensors" or "sensing" so that it would be easier for potential readers to find your study. 

(2) Please include a "Limitations" section to discuss what were the challenges faced, and how your team overcame those challenges. This would be very beneficial to the readers as they would be able to learn from your expert knowledge.

(3) To improve the impact and readership of your manuscript, the authors need to clearly articulate in the Abstract and in the Introduction sections about the uniqueness or novelty of this article, and why or how it is different from other similar articles. Can the authors please kindly elaborate more about how this study is relevant to "sensors" since it was submitted for publication in the journal entitled "Sensors"?

(4) Please substantially expand your review work, and cite more of the journal papers published by MDPI.

(5) All of the references cited are not yet properly formatted according to MDPI's guidelines. For example, the DOIs of all the journal papers cited are not included yet. For the references, instead of formatting "by-hand", please kindly consider using the free Zotero software (https://www.zotero.org/), and select "Multidisciplinary Digital Publishing Institute" as the citation format, since there are currently 47 citations in your manuscript, and there may probably be more once you have revised the manuscript.

Thank you.

Reviewer 2 Report

The manuscript deals with a distributed machine learning framework in distributed systems. Overall, the manuscript is well-organized and -presented. The abstract is well defined but numeric summaries can be added. In the Introduction, the core concept is well-presented with illustrative figures. Contributions to the paper are also well-presented. Section 2 starts with a subsection without guidance. Scientific representation is required near equations. In Section 3, the system model is well-defined, but a more formal definition of problem formulation can be expected. Section 4 describes the core algorithms, but it starts with a specific algorithm. I recommend adding a high-level overview (with a figure if possible) from a bird's-eye view. Section 5 supports the proposed method with formal proof. Section 6 also starts with a subsection followed by a subsubsection. The sentence starts with "Note all" is out of the blue. Section 6.1 lacks hardware descriptions or simulation environments information. Figure 3 is hard to recognize due to its small font size. For the result part, I recommend inserting a discussion about the authors' insights and implications, not just describing the result itself. Furthermore, a comparative analysis with state-of-the-art studies is required (critical issue).
Overall, the manuscript requires major revisions.

Round 2

Reviewer 2 Report

The authors revised the manuscript based on the previous review.

Thus, I recommend the manuscript for publication.